# Disturbance-based Discretization, Differentiable IDS Channel, and an IDS-Correcting Code for DNA Storage

## Abstract

Insertion, deletion, and substitution (IDS) error-correcting codes have garnered increased attention with recent advancements in DNA storage technology. However, a universal method for designing tailored IDS-correcting codes across varying channel settings remains under-explored. We present an autoencoder-based approach, THEA-code, aimed at efficiently generating IDS-correcting codes for complex IDS channels. In the work, a disturbance-based discretization is proposed to discretize the features of the autoencoder, and a simulated differentiable IDS channel is developed as a differentiable alternative for IDS operations. These innovations facilitate the successful convergence of the autoencoder, producing channel-customized IDS-correcting codes that demonstrate commendable performance across complex IDS channels, particularly in the realistic DNA storage channel.

## 1. Introduction

DNA storage, a method that uses the synthesis and sequencing of DNA molecules for information storage and retrieval, has attracted significant attention (Church et al., 2012; Goldman et al., 2013; Grass et al., 2015; Erlich & Zielinski, 2017; Organick et al., 2018; Dong et al., 2020; Chen et al., 2021; El-Shaikh et al., 2022; Welzel et al., 2023).

Due to the involvement of biochemical procedures, the DNA storage pipeline can be viewed as an insertions, deletions, or substitutions (IDS) channel (Blawat et al., 2016) over 4-ary sequences with the alphabet $\{A, T, G, C\}$. Consequently, an IDS-correcting encoding/decoding method plays a key role in DNA storage.

However, despite the existence of excellent combinatorial

IDS-correcting codes (Varshamov & Tenenholtz, 1965; Levenshtein, 1965; Sloane, 2000; Mitzenmacher, 2009; Cai et al., 2021; Gabrys et al., 2023; Bar-Lev et al., 2023), applying them in DNA storage remains challenging. The IDS channel in DNA storage is more complex than those studied in previous works, with factors such as inhomogeneous error probabilities across error types, base indices, and even sequence patterns (Hirao et al., 1992; Press et al., 2020; Blawat et al., 2016; Cai et al., 2021; Hamoum et al., 2021). Additionally, most of the aforementioned combinatorial codes focus on correcting either a single error or a burst of errors, whereas multiple independent errors within the same DNA sequence are common in DNA storage.

Given the complexity of the IDS channel, we leverage the universality of deep learning methods by employing a heuristic end-to-end autoencoder (Baldi, 2012) as the foundation for an IDS-correcting code. This approach enables researchers to train customized codes tailored to various IDS channels through a unified training procedure, rather than manually designing specific combinatorial codes for each IDS channel setting, many of which remain unexplored.

To realize this approach, two novel techniques are developed, which we believe offer greater contributions to the communities than the code itself.

Firstly, the discretization effect of applying disturbance in a non-generative model is investigated in this work. It is observed that introducing disturbance to the logistic feature forces the non-generative model to reduce the disturbance caused indeterminacy by producing more confident logits, thereby achieving discretization. This aligns with the discrete codewords of an error-correcting code (ECC) in this work, and provides an alternative approach for bridging the gap between continuous models and discrete applications.

Secondly, a differentiable IDS channel using a transformer-based model (Vaswani et al., 2017) is developed. The non-differentiable nature of IDS operations presents a key challenge for deploying deep learning models that rely on gradient descent training. To tackle this, a model is trained in advance to mimic the IDS operations according to a given error profile. It can serve as a plug-in module for the IDS channel and is backpropagable within the network. This dif-

[1]Anonymous Institution, Anonymous City, Anonymous Region, Anonymous Country. Correspondence to: Anonymous Author <anon.email@domain.com>.

Preliminary work. Under review by the International Conference on Machine Learning (ICML). Do not distribute.

ferentiable IDS channel has the potential to act as a general module for addressing IDS or DNA-related problems using deep learning methods. For instance, researchers could build generative models on this module to simulate the biochemical processes involved in manipulating biosequences.

Overall, this work implements a heuristic end-to-end autoencoder as an IDS-correcting code, referred to as THEA-Code. The encoder maps the source DNA sequence into a longer codeword sequence. After introducing IDS errors to the codeword, a decoder network is employed to reconstruct the original source sequence from the codeword. During the training of this autoencoder, disturbance-based discretization is applied to the codeword sequence to produce one-hot-like vectors, and the differentiable IDS channel serves as a substitute for conventional IDS channel, enabling gradient backpropagation.

To the best of our knowledge, this work presents the first end-to-end autoencoder solution for an IDS-correcting code. It introduces the disturbance-based discretization, and proposes the first differentiable IDS channel. It is also the first universal method for designing tailored IDS-correcting codes across varying channel settings. Experiments across multiple complex IDS channels, particularly in the realistic DNA storage channel, demonstrate the effectiveness of the proposed THEA-Code.

## 2. Related Works

Many established IDS-correcting codes are rooted in the Varshamov-Tenengolts (VT) code (Varshamov & Tenengolts, 1965; Levenshtein, 1965), including (Calabi & Hartnett, 1969; Tanaka & Kasai, 1976; Sloane, 2000; Cai et al., 2021; Gabrys et al., 2023). These codes often rely on rigorous mathematical deduction and provide firm proofs for their coding schemes. However, the stringent hypotheses they use tend to restrict their practical applications. Heuristic IDS-correcting codes for DNA storage, such as those proposed in (Pfister & Tal, 2021; Yan et al., 2022; Maarouf et al., 2022; Welzel et al., 2023), usually incorporate synchronization markers (Sellers, 1962; Srinivasavaradhan et al., 2021; Haeupler & Shahrasbi, 2021), watermarks (Davey & Mackay, 2001), or positional information (Press et al., 2020) within their encoded sequences. Recently, directly correcting errors in retrieved DNA reads without sequence reconstruction has been investigated, demonstrating promising performance (Welter et al., 2024).

In recent years, deep learning methods have found increasing applications in coding theory (Ibnkhala, 2000; Simeone, 2018; Akrout et al., 2023). Several architectures have been employed as decoders or sub-modules of conventional codes on the AWGN channel. In (Cammerer et al., 2017), the authors applied neural networks to replace sub-blocks in the

conventional iterative decoding algorithm for polar codes. Recurrent neural networks (RNN) were used for decoding convolutional and turbo codes (Kim et al., 2018). Both RNNs and transformer-based models have served as belief propagation decoders for linear codes (Nachmani et al., 2018; Choukroun & Wolf, 2022; 2023; 2024b;a;c). Hypergraph networks were also utilized as decoders for block codes in (Nachmani & Wolf, 2019). Despite these advancements, end-to-end deep learning solutions remain relatively less explored. As mentioned in (Jiang et al., 2019), direct applications of multi-layer perceptron (MLP) and convolutional neural network (CNN) are not comparable to conventional methods. To address this, the authors in (Jiang et al., 2019) used deep models to replace sub-modules of a turbo code skeleton, and trained an end-to-end encoder-decoder model. Similarly, in (Makkuva et al., 2021), neural networks were employed to replace the Plotkin mapping for the Reed-Muller code. Both of these works inherit frameworks from conventional codes and utilize neural networks as replacements for key modules. In (Balevi & Andrews, 2020), researchers proposed an autoencoder-based inner code with one-bit quantization for the AWGN channel. Confronting challenges arising from quantization, they utilized interleaved training on the encoder and decoder.

## 3. Disturbance-based Discretization

In this work, it is observed that introducing disturbance to the categorical distribution feature produced by a non-generative model causes the feature to resemble a one-hot vector. Intuitively, the non-generative model may attempt to reduce the indeterminacy introduced by the distrubance by generating more confident logits.

Let $\boldsymbol{x}$ be the logits that produce the probabilities $\boldsymbol{\pi} = \{\pi_1, \pi_2, \ldots, \pi_k\}$ via the softmax function,

$$\pi_i = \frac{\exp x_i}{\sum_{j=1}^{k} \exp x_j}, \quad i = 1, 2, \ldots, k. \quad (1)$$

In this work, the non-generative disturbance is introduced to $\boldsymbol{\pi}$ by sampling from the Gumbel distribution (Gumbel, 1935). It follows the same formula as the Gumbel-Softmax, which has been widely used in generative models for generating samples (Jang et al., 2017; Maddison et al., 2017; Huijben et al., 2023). Specifically, the non-generative disturbance is applied to $\boldsymbol{x}$ using the following formula:

$$\mathrm{GS}(\boldsymbol{x})_i = \frac{\exp\left((x_i + g_i)/\tau\right)}{\sum_{j=1}^{k} \exp\left((x_j + g_j)/\tau\right)}, \quad i = 1, 2, \ldots, k, \quad (2)$$

where $g_1, g_2, \ldots, g_k$ are i.i.d. samples drawn from the Gumbel distribution $G(0, 1)$ and $\tau$ is the temperature that controls the entropy.

Applying $\mathrm{GS}(\boldsymbol{x})$ in a non-generative model is found to

induce the model to produce more confident logits $x$ and, consequently, probabilities $\pi$ that resemble one-hot vectors, as stated in Theorem 3.1.

**Theorem 3.1.** *By introducing disturbance to a non-generative autoencoder's feature logits $x$ via $\mathrm{GS}(x)$, the autoencoder, upon non-trivial convergence, produces confident logits $x$, resulting in one-hot-like probabilities $\pi$.*

*Brief proof:* For simplicity, the binary case is considered as an example, with the temperature set to $\tau = 1$. Let $x = (x_1, x_2)$ represent the logits outputted by the upstream model, and let $y = \mathrm{GS}(x)$ denote the Gumbel-Softmax of $x$, where

$$y_i = \frac{\exp(x_i + g_i)}{\exp(x_1 + g_1) + \exp(x_2 + g_2)}, \quad i = 1, 2. \quad (3)$$

Let $\mathcal{L} = f(y_1, y_2)$ be the optimization target of the model, which represents the composite function of the downstream model and the loss function.

The partial derivative of the optimization target with respect to $x_1$ is calculated using the chain rule:

$$\frac{\partial \mathcal{L}}{\partial x_1} = \frac{\partial f}{\partial y_1}\frac{\partial y_1}{\partial x_1} + \frac{\partial f}{\partial y_2}\frac{\partial y_2}{\partial x_1} = y_1 y_2 \left( \frac{\partial f}{\partial y_1} - \frac{\partial f}{\partial y_2} \right). \quad (4)$$

The calculations for $x_2, y_2$ are analogous to those for $x_1, y_1$ and are omitted here and in the following text. It is known that a model converges to a local minimum has a zero gradient, thus according to Equation (4), when the model converges, either $y_1, y_2$ or $\left| \frac{\partial f}{\partial y_1} - \frac{\partial f}{\partial y_2} \right|$ should be zero or less than a minimal value $\epsilon$.

Consider the case where $\left| \frac{\partial f}{\partial y_1} - \frac{\partial f}{\partial y_2} \right| < \epsilon$. Noting that $y_2 = 1 - y_1$, the partial derivative of $\mathcal{L}$ with respect to $y_1$ is given by

$$\left| \frac{\partial \mathcal{L}}{\partial y_1} \right| = \left| \frac{\partial f}{\partial y_1} + \frac{\partial f}{\partial y_2}\frac{\partial y_2}{\partial y_1} \right| = \left| \frac{\partial f}{\partial y_1} - \frac{\partial f}{\partial y_2} \right| < \epsilon. \quad (5)$$

Since $y_1$ is calculated by sampling a random variable as described in Equation (3), either $y_1$ remains constant with respect to different $g_i$, or Equation (5) holds for a variable $y_i$, in which case the downstream model degenerates into a trivial model, as its optimization target becomes insensitive to different inputs. The partial derivative of $y_1$ with respect to $g_1$ is

$$\frac{\partial y_1}{\partial g_1} = y_1 y_2. \quad (6)$$

If $y_1$ is not sensitive to $g_1$, either $y_1$ or $y_2$ should be zero or less than a minimal value $\epsilon$.

All the above cases indicate that the converged model should have either $y_1$ or $y_2$ less than $\epsilon$. Taking $y_1 < \epsilon_1$ as an example, it can be reformulated as

$$\frac{1}{y_1} = 1 + \exp(x_2 - x_1 + g_2 - g_1) > M_1, \quad (7)$$

where $M_1 = 1/\epsilon_1$. Since $g_1, g_2$ independently follow the Gumbel distribution $G(0, 1)$, whose probability density function (PDF) is

$$f_{G(0,1)}(x) = \exp(-x)\exp(-\exp(-x)), \quad (8)$$

the distribution of $g_2 - g_1$ can be calculated by convolution, resulting in a logistic distribution $\mathrm{Logistic}(0, 1)$ with PDF

$$f_{\mathrm{Logistic}(0,1)}(x) = \frac{\exp(-x)}{(1 + \exp(-x))^2}. \quad (9)$$

Thus, the probability of $1/y_1$ being greater than $M_1 = 1/\epsilon_1$ is

$$P_{g_2 - g_1}\left( \frac{1}{y_1} > M_1 \right) = 1 - \frac{M_1 - 1}{\exp(x_2 - x_1) + (M_1 - 1)}. \quad (10)$$

Letting this probability be greater than $1 - \epsilon_2$, the $x_1, x_2$ should follow the restriction

$$\exp(x_2 - x_1) > (M_1 - 1)(M_2 - 1), \quad (11)$$

where $M_2 = 1/\epsilon_2$. This indicates that the upstream network should produce confident logits $x$, as applying softmax to $x$ results in

$$\pi_1 = \frac{\exp x_1}{\exp x_1 + \exp x_2} < \frac{1}{(M_1 - 1)(M_2 - 1) + 1}$$
$$< \frac{2}{M_1 M_2} = 2\epsilon_1 \epsilon_2, \quad (12)$$

when $\epsilon_1 + \epsilon_2 < 0.5$. $\square$

Based on the above analysis, it can be inferred that a converged model using the disturbance of Equation (2) on its feature $x$ instead of the vanilla softmax will constrain the logits $x$ to produce one-hot-like probability vectors.

## 4. Differentiable IDS Channel on $3$-Simplex $\Delta^3$

It is evident that the operations of insertion and deletion are not differentiable. Consequently, a conventional IDS channel, which modifies a sequence by directly applying IDS operations, hinders gradient propagation and cannot be seamlessly integrated into deep learning-based methods.

Leveraging the logical capabilities inherent in transformer-based models, a sequence-to-sequence model is employed to simulate the conventional IDS channel. Built on deep models, this simulated IDS channel is differentiable. In the following discussion, we use the notation $\mathrm{CIDS}(\cdot, \cdot)$ to represent the Conventional IDS channel, and $\mathrm{DIDS}(\cdot, \cdot; \theta)$ for the simulated Differentiable IDS channel. The simulated channel is trained independently before being integrated into the autoencoder, whose learned parameters remain fixed during the optimization of the autoencoder.

As the model utilizes probability vectors rather than discrete letters, we need to promote conventional IDS operations onto the 3-simplex $\Delta^3$, where $\Delta^3$ is defined as the collection 4-dimentional probability vectors

$$\Delta^3 = \{\boldsymbol{\pi}\,|\,\pi_i \geq 0, \sum_{i=1}^{4} \pi_i = 1, i = 1, 2, 3, 4\}. \qquad (13)$$

For a sequence of probability vectors $C = (\boldsymbol{\pi}_1, \boldsymbol{\pi}_2, \ldots, \boldsymbol{\pi}_k)$, where each $\boldsymbol{\pi}_i$ is an element from the simplex $\Delta^3$, the IDS operations are promoted as follows.

Insertion at index $i$ involves adding a one-hot vector representing the inserted symbol from the alphabet $\{A, T, G, C\}$ before index $i$. Deletion at index $i$ simply removes the vector $\boldsymbol{\pi}_i$ from $C$. For substitution, the probability vector $\boldsymbol{\pi}_i$ is rolled by corresponding offsets for the three types of substitutions (type-$1, 2, 3$). For example, applying a type-$1$ substitution at index $i$ rolls the original vector $\boldsymbol{\pi}_i = (\pi_{i1}, \pi_{i2}, \pi_{i3}, \pi_{i4})$ into $(\pi_{i4}, \pi_{i1}, \pi_{i2}, \pi_{i3})$. It is straightforward to verify that the promoted IDS operations degenerate to standard IDS operations when the probability vectors are constrained to a one-hot representation.

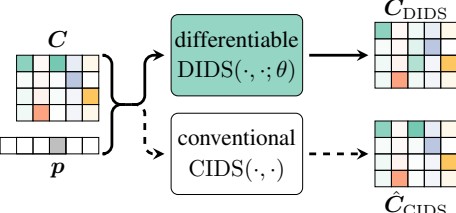

Figure 1: The differentiable IDS channel. The $\hat{C}_{\text{DIDS}}$ and $\hat{C}_{\text{CIDS}}$ are generated by the differentiable and conventional IDS channels, respectively. Optimizing the difference between $\hat{C}_{\text{DIDS}}$ and $\hat{C}_{\text{CIDS}}$ trains the differentiable channel.

As illustrated in Figure 1, both the conventional IDS channel CIDS and the simulated IDS channel DIDS take the sequence $C$ of probability vectors and an error profile $p$ as their inputs. The error profile consists of a sequence of letters that record the types of errors encountered while processing $C$. Complicated IDS channels can be deduced by specifying the rules for generating error profiles. The probability sequence $C$ is expected to be modified by the simulated IDS channel to $\hat{C}_{\text{DIDS}} = \text{DIDS}(C, p; \theta)$ according to the error profile $p$ in the upper stream of Figure 1. In the lower stream, the sequence $C$ is modified as $\hat{C}_{\text{CIDS}} = \text{CIDS}(C, p)$ with respect to the error profile $p$ using the previously defined promoted IDS operations.

To train the model $\text{DIDS}(\cdot, \cdot; \theta)$, the Kullback–Leibler divergence (Kullback, 1997) of $\hat{C}_{\text{DIDS}}$ from $\hat{C}_{\text{CIDS}}$ can be utilized as the optimization target

$$\mathcal{L}_{\text{KLD}}(\hat{C}_{\text{DIDS}}, \hat{C}_{\text{CIDS}}) = \frac{1}{k} \sum_i \hat{\boldsymbol{\pi}}_{i\text{CIDS}}^T \log \frac{\hat{\boldsymbol{\pi}}_{i\text{CIDS}}}{\hat{\boldsymbol{\pi}}_{i\text{DIDS}}}. \qquad (14)$$

By optimizing Equation (14) on randomly generated probability vector sequences $C$ and error profiles $p$, the parameters $\theta$ of the differentiable IDS channel are trained to $\hat{\theta}$. Following this, the model $\text{DIDS}(\cdot, \cdot; \hat{\theta})$ simulates the conventional IDS channel $\text{CIDS}(\cdot, \cdot)$. The significance of such an IDS channel lies in its differentiability. Once optimized independently, the parameters of the IDS channel are fixed for downstream applications. In the following text, we use $\text{DIDS}(\cdot, \cdot)$ to refer to the trained IDS channel for simplicity.

In practice, the differentiable IDS channel is implemented as a sequence-to-sequence model, employing one-layer transformers for both its encoder and decoder[1]. The model takes a padded vector sequence and error profile, whose embeddings are concatenated along the feature dimension as its input. To generate the output, that represents the sequence with errors, learnable position embedding vectors are utilized as the queries (omitted from Figure 1).

## 5. THEA-Code

### 5.1. Framework

The flowchart of the proposed code is illustrated in Figure 2. Based on the principles of DNA storage, which synthesizes DNA molecules of fixed length, the proposed model is designed to handle source sequences and codewords of constant lengths. Essentially, the proposed method encodes source sequences into codewords; the IDS channel introduces IDS errors to these codewords; and a decoder is employed to reconstruct the sink sequences according to the corrupted codewords.

Let $f_{\text{en}}(\cdot; \phi)$ denote the encoder, where $\phi$ represents the encoder's parameters. The source sequence $s$ is first encoded into the codeword $c = f_{\text{en}}(s; \phi)$ by the encoder[2], where the codeword $c$ is obtained using Equation (2) during the training phase and $\text{argmax}$ during the testing phase. Next, a random error profile $p$ is generated, which records the positions and types of errors that will occur on codeword $c$. Given the error profile $p$, the codeword $c$ is transformed into the corrupted codeword $\hat{c} = \text{DIDS}(c, p; \hat{\theta})$ by the simulated differentiable IDS channel, implemented as a sequence-to-sequence model with trained parameters $\hat{\theta}$. Finally, a de-

---

[1]Here, the encoder and decoder refer specifically to the modules of the sequence-to-sequence model, not the modules of the autoencoder. We trust that readers will be able to distinguish between them based on the context.

[2]For simplicity, we do not distinguish between notations for sequences represented as letters, one-hot vectors, or probability vectors in the following text.

coder $f_{\text{de}}(\cdot; \psi)$ with parameters $\psi$ decodes the corrupted codeword $\hat{c}$ back into the sink sequence $\hat{s} = f_{\text{de}}(\hat{c}; \psi)$.

Following this pipeline, a natural optimization target is the cross-entropy loss

$$\mathcal{L}_{\text{CE}}(\hat{s}, s) = -\sum_i \sum_j \mathbb{1}_{j=s_i} \log \hat{s}_{ij}, \qquad (15)$$

which evaluates the reconstruction disparity of the source sequence $s$ by the sink sequence $\hat{s}$.

However, merely optimizing such a loss function will not yield the desired outcomes. While the encoder and decoder of an autoencoder typically collaborate on a unified task in most applications, in this work, we expect them to follow distinct underlying logic. Particularly, when imposing constraints to enforce greater discreteness in the codeword, the joint training of the encoder and decoder resembles a chicken-and-egg dilemma, where the optimization of each relies on the other during the training phase.

### 5.2. Auxiliary reconstruction of source sequence by the encoder

To address the aforementioned issue, we introduce a supplementary task exclusively for the encoder, aimed at initializing it with some foundational logical capabilities. Inspired by the systematic code which embed the input message within the codeword, a straightforward task for the encoder is to replicate the input sequence at the output, ensuring that the model preserves all information from its input without reduction. With this in mind, we incorporate a reconstruction task into the encoder's training process.

In practice, the encoder is designed to output a longer sequence, which is subsequently split into two parts: the codeword representation $c$ and a auxiliary reconstruction $r$ of the input source sequence, as shown in Figure 2. The auxiliary reconstruction loss is calculated using the cross-entropy loss as

$$\mathcal{L}_{\text{Aux}}(r, s) = -\sum_i \sum_j \mathbb{1}_{j=s_i} \log r_{ij}, \qquad (16)$$

which quantifies the difference between the reconstruction $r$ and the input sequence $s$.

Considering that the auxiliary loss may not have negative effects on the encoder for its simple logic, we don't use a separate training stage for optimizing the $\mathcal{L}_{\text{Aux}}$. The auxiliary loss defined in Equation (16) is incorporated into the overall loss function and applied consistently throughout the entire training phase.

### 5.3. The encoder and decoder

In this approach, both the encoder and decoder are implemented using transformer-based sequence-to-sequence mod-els. Each consists of (3+3)-layer transformers with sinusoidal positional encoding. The embedding of the DNA bases is implemented through a fully connected layer without bias to ensure compatibility with probability vectors. Learnable position index embeddings are employed to query the outputs.

### 5.4. Training phase

The training process is divided into two phases. Firstly, the differentiable IDS channel is fully trained by optimizing

$$\hat{\theta} = \arg\min_\theta \mathcal{L}_{\text{KLD}}(\hat{C}_{\text{DIDS}}, \hat{C}_{\text{CIDS}}) \qquad (17)$$

on randomly generated codewords $c$ and profiles $p$. Once the differentiable IDS channel is trained, its parameters are fixed. The remaining components of the autoencoder are then trained by optimizing a weighted sum of Equation (15) and Equation (16),

$$\hat{\phi}, \hat{\psi} = \arg\min_{\phi, \psi} \mathcal{L}_{\text{CE}}(\hat{s}, s) + \mu \mathcal{L}_{\text{Aux}}(r, s), \qquad (18)$$

where $\mu$ is a hyperparameter representing the weight of the auxiliary reconstruction loss. The autoencoder is trained on randomly generated input sequences $s$ and profiles $p$.

### 5.5. Testing phase

In the testing phase, the differentiable IDS channel is replaced with the conventional IDS channel. The process begins with the encoder mapping the source sequence $s$ to the codeword $c$ in the form of probability vectors. An $\texttt{argmax}$ function is then applied to convert $c$ into a discrete letter sequence, removing any extra information from the probability vectors. Next, the conventional IDS operations are performed on $\hat{c} = \text{CIDS}(c, p)$ according to a randomly generated error profile $p$. The one-hot representation of $\hat{c}$ is then passed into the decoder, which reconstructs the sink sequence $\hat{s}$. Finally, metrics are computed to measure the differences between the original source sequence $s$ and the reconstructed sink sequence $\hat{s}$, providing an evaluation of the method's performance.

Since the sequences are randomly generated from an enormous pool of possible terms, the training and testing sets are separated using different random seeds. For example, in the context of this work, the source sequence is a 100-long 4-ary sequence, providing $1.6 \times 10^{60}$ possible sequences. Given this vast space, sets of randomly generated sequences using different seeds are unlikely to overlap.

## 6. Experiments and Ablation Study

Commonly used methods for synthesizing DNA molecules in DNA storage pipelines typically yield sequences of

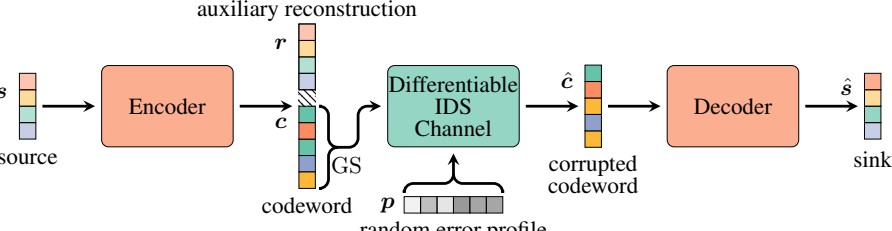

Figure 2: The flowchart of THEA-Code, including the encoder, the pretrained IDS channel, and the decoder. All of these modules are implemented using transformer-based models.

lengths ranging from 100 to 200 (Welter et al., 2024). In this study, we choose the number 150 as the codeword length, aligning with these established practices. Unless explicitly stated otherwise, all the following experiments adhere to the default setting: source sequence length $\ell_s = 100$, codeword length $\ell_c = 150$, auxiliary loss weight $\mu = 1$, and the error profile is generated with a $1\%$ probability of errors occurring at each position, with insertion, deletion, and substitution errors equally likely.

To evaluate performance, the nucleobase error rate (NER) is employed as a metric, analogous to the bit error rate (BER), but replacing bits with nucleobases. For a DNA sequence $s$ and its decoded counterpart $\hat{s}$, the NER is defined as

$$\text{NER}(s, \hat{s}) = \frac{\#\{s_i \neq \hat{s}_i\}}{\#\{s_i\}}. \qquad (19)$$

The NER represents the proportion of nucleobase errors corresponding to base substitutions in the source DNA sequence. It's worth noting that these errors can be post-corrected using a mature conventional outer code.

The source code is uploaded at https://anonymous.4open.science/r/THEACode, and will be made publicly accessible upon the manuscript's publication.

### 6.1. Effects of the disturbance-based discretization

The ablation study on utilizing the disturbance-based discretization was conducted analyzing the discreteness of the codewords. During training, the entropy of the codewords

$$H(\boldsymbol{\pi}) = -\sum_{i=1}^{k} \pi_i \log \pi_i \qquad (20)$$

was recorded. This entropy measures the level of discreteness in the codewords. Lower entropy implies a distribution that is closer to a one-hot style probability vector, which indicates greater discreteness. In addition to entropy, two other metrics were also recorded, as they are the reconstruction loss $\mathcal{L}_{\text{CE}}$ and the NER. The results, plotted in Figure 3, compare the default disturbance setting (Gumbel-Softmax) against a vanilla softmax approach.

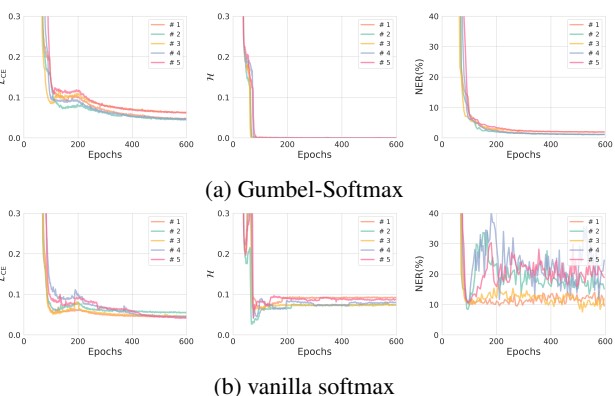

(a) Gumbel-Softmax

(b) vanilla softmax

Figure 3: The reconstruction loss, codeword entropy, and validation NER comparing the Gumbel-Softmax setting against a vanilla softmax approach. 5 runs were recorded.

The first column of Figure 3 indicates that using disturbance marginally increases the reconstruction loss $\mathcal{L}_{\text{CE}}$ in the continuous mode, which is expected since Gumbel-Softmax introduces additional noise into the system. When comparing the average entropy $\mathcal{H}$ of the learned codeword, applying disturbance-based discretization significantly reduces the entropy, suggesting that the codewords behave more like one-hot vectors. The NER is calculated in the discrete mode by replacing the softmax with an argmax operation on the codewords. The third column clearly shows that when codewords are closer to a one-hot style, the model is more consistent between the continuous and discrete modes, leading to better performance during the testing phase.

The hyperparameter optimization of the Gumbel-Softmax temperature $\tau$ is presented in Appendix B.

### 6.2. Performance with different channel settings

The code rate is the proportion of non-redundant data in the codeword, calculated by dividing the source length $\ell_s$ by the codeword length $\ell_c$. We explored variable source lengths $\ell_s$ while keeping the codeword length $\ell_c = 150$ fixed. The results in Table 1 reveal a trend that the NER increases from $0.09\%$ to $2.81\%$ as the code rate increases from $0.33$ to $0.83$.

Table 1: The testing NER for different source lengths $\ell_s$.

| $\ell_s$ | 50 | 75 | 100 | 125 |
|---|---|---|---|---|
| code rate | 0.33 | 0.50 | 0.67 | 0.83 |
| NER(%) | 0.09 | 0.46 | 1.06 | 2.81 |

By applying an outer conventional ECC to address the remaining NER, which is a common technique in DNA storage (Press et al., 2020; Pfister & Tal, 2021; Yan et al., 2022; Welzel et al., 2023), a complete solution for DNA storage is achieved. Here, the IDS-correcting code is focused.

By controlling the generation process of the error profile $p$ for different channel settings, we can evaluate whether THEA-Code learns channels' attributes and produces customized codes based on the models' performance.

**Results on IDS channels with position related errors.**
Along with the default setting, where error rates are position-insensitive (denoted as Hom), two other IDS channels parameterized by ascending (Asc) and descending (Des) error rates along the sequence are considered[3]. The Asc channel has error rates increasing from 0% to 2% along the sequence, with the average error rate matching that of the default setting Hom. The Des channel follows a similar pattern but has decreasing error rates along the sequence.

To verify that the proposed method customizes codes for different channels, cross-channel testing was conducted, with the results shown in Table 2. The numbers in the matrix represent the NER of a model trained with the channel of the row and tested on the channel of the column.

Table 2: The testing NER across different channels. Each entry represents the NER of a model trained (resp. tested) on the channel specified by the row (resp. column) header.

| NER(%) | Asc | Hom | Des |
|---|---|---|---|
| Asc | **0.90** | 1.46 | 2.09 |
| Hom | 1.03 | **1.15** | 1.30 |
| Des | 1.72 | 1.32 | **1.01** |

The diagonal of Table 2 shows the results of the model trained and tested with a consistent channel, suggesting that the learned THEA-Code exhibits varying performance depending on the specific channel configuration. The columns of Table 2 suggest that, for each testing channel, models trained with the channel configuration consistently achieve the best performance among the three channel settings. Considering the Hom channel is a midway setting between Asc and Des, the first and third columns (and rows) show that the more dissimilar the training and testing channels are, the

---

[3]These settings simplify DNA storage channels, as a DNA sequence is marked with a 3' end and a 5' end. Some researchers believe that the error rate accumulates towards the sequence end during synthesis (Meiser et al., 2020).

worse the model's performance becomes, even though the overall error rates are the same across the three channels. These findings verify that the deep learning-based method effectively customizes codes for specific channels, which could advance IDS-correcting code design into a more fine-grained area.

**Results on IDS channels with various IDS error rates.**
IDS channels with larger error probabilities were also tested. The experiments were extended to include channels with error probabilities in $\{0.5\%, 1\%, 2\%, 4\%, 8\%, 16\%\}$, with results listed in Table 3.

It is suggested that models trained on channels with higher error probabilities exhibit compatibility with channels with lower error probabilities. In most cases, models trained and tested on similar channels achieve better performance.

Table 3: The testing NER across different IDS error probabilities. The row and column headers correspond to channels configured with respective probabilities of errors. Each entry represents the NER of a model trained (resp. tested) on the channel specified by the row (resp. column) header.

| NER(%) | 0.5% | 1% | 2% | 4% | 8% | 16% |
|---|---|---|---|---|---|---|
| 0.5% | **0.68** | 1.59 | 4.26 | 11.67 | 26.87 | 45.61 |
| 1% | 0.52 | **1.15** | 2.9 | 8.12 | 21.19 | 41.03 |
| 2% | 0.67 | 1.43 | **3.16** | 7.79 | 18.7 | 36.89 |
| 4% | 1.25 | 1.76 | 2.88 | **5.53** | 12.39 | 28.31 |
| 8% | 2.74 | 3.24 | 4.30 | 6.62 | **12.2** | 25.41 |
| 16% | 11.57 | 11.93 | 12.61 | 14.4 | 17.22 | **25.51** |

**Results on realistic IDS channels.** We also conducted experiments using IDS channels that more closely resemble realistic IDS channels in DNA storage. A memory channel was proposed in (Hamoum et al., 2021), relying on statistical data obtained via Nanopore sequencing. It models the IDS errors based on the $k$-mers of sequences and adjacent edits. In this work, we utilize the publicly released trained memory channel from (Hamoum et al., 2021), filtering out apperent outlier sequences with Levenshtein distance greater than 20. This simulated channel is referred to as MemSim.

In practice, a DNA sequence $c$ is input into MemSim to produce the output sequence $\hat{c}$ from the channel. By comparing $c$ and $\hat{c}$, an error profile $p$ is inferred. Using the sequence $c$ and the error profile $p$ in the procedure depicted in Figure 2, an IDS-correcting code customized for MemSim is trained.

For comparison, two simple channels, partially aligned with MemSim, were also considered. The overall IDS error rate for MemSim is 10.36%, with the proportions of insertion, deletion, and substitution being 1.66%, 5.31%, and 3.38%, respectively. We refer to the context-free channel with these specific error proportions as channel C253. Channel C111 is defined as having the same overall IDS error rate 10.36%, but with equal proportions of insertion, deletion, and substi-

Table 4: The testing NER across different channels including C111, C253, and MemSim, under varying code rates. Each entry represents the NER of a model trained (resp. tested) on the channel specified by the row (resp. column) header.

| | $r = 0.33$ | | | $r = 0.50$ | | | $r = 0.67$ | | |
| --- | --- | --- | --- | --- | --- | --- | --- | --- | --- |
| NER(%) | C111 | C253 | MemSim | C111 | C253 | MemSim | C111 | C253 | MemSim |
| C111 | **2.28** | 3.02 | 15.9 | **7.60** | 8.77 | 24.85 | **15.19** | 16.96 | 34.46 |
| C253 | 2.73 | **2.93** | 17.3 | 9.15 | **9.13** | 25.09 | 16.87 | **16.90** | 32.86 |
| MemSim | 5.60 | 6.64 | **1.55** | 14.78 | 16.62 | **6.11** | 24.89 | 25.91 | **12.02** |

Table 5: The testing error rates compared with established code through channels including C111, C253, and MemSim, under varying code rates.

| | $r = 0.33$ | | | $r = 0.50$ | | | $r = 0.67$ | | |
| --- | --- | --- | --- | --- | --- | --- | --- | --- | --- |
| | C111 | C253 | MemSim | C111 | C253 | MemSim | C111 | C253 | MemSim |
| Cai | 17.01 | 17.52 | 72.74 | 29.00 | 29.57 | 74.40 | 40.12 | 42.62 | 73.90 |
| DNA-LM | 32.24 | 37.33 | 60.13 | 45.32 | 51.13 | 64.27 | 56.34 | 60.22 | 68.72 |
| HEDGES | 3.21 | 4.56 | 29.42 | 27.22 | 27.79 | 99.56 | 54.35 | 55.66 | 99.62 |
| THEA-Code | **2.28** | **2.93** | **1.55** | **7.60** | **9.13** | **6.11** | **15.19** | **16.90** | **12.02** |

tution. It is evident that MemSim is the closest approximation to a realistic channel, followed by C253, while C111 deviates the most from a realistic channel.

The results across channels, including C111, C253, and MemSim, are presented in Table 4. The results suggest that THEA-Code performs better when the model is trained on the same channel used for testing. Specifically, for the realistic channel, codes trained on the simpler channels C253 and C111 fail to deliver satisfactory results. Overall, THEA-Code trained and tested with MemSim achieves the best results, demonstrating that the proposed model significantly benefits from customizing the code for the realistic channel.

### 6.3. Comparison experiments

Table 6: The testing error rates compared with different established codes, through the default 1% IDS channel.

| code rate | 0.33 | 0.50 | 0.6 | 0.67 | 0.75 | 0.83 |
| --- | --- | --- | --- | --- | --- | --- |
| Cai | 0.44 | 1.00 | - | 2.53 | - | 8.65 |
| DNA-LM | 0.55 | 1.03 | - | 2.29 | - | 7.43 |
| HEDGES | 0.28 | **0.25** | 0.65 | - | 3.43 | - |
| THEA-Code | **0.09** | 0.46 | - | **1.06** | - | **2.81** |

Comparison experiments were conducted against prior works include: the combinatorial code from (Cai et al., 2021), the segmented code method DNA-LM from (Yan et al., 2022), and the efficient heuristic method HEDGES from (Press et al., 2020).

Such methods are typically designed to operate under discrete, fixed configurations, making it challenging to align them within the same setting. We made every effort to align these methods, and present a subset of the comparison results in Table 6, which is tested through the default 1% error channel. Detailed configurations and results across multiple channels are provided in Appendix A.

Table 6 demonstrates the effectiveness of the proposed method. The performance of THEA-Code and HEDGES outperform the other methods by a large margin. At lower code rates, THEA-Code achieves a comparable error rate to HEDGES. At higher code rates, the proposed method outperforms HEDGES, achieving a lower error rate at a higher code rate, specifically $2.81\%$ error rate at $0.83$ code rate for THEA-Code v.s. $3.43\%$ error rate at $0.75$ code rate for HEDGES.

**Comparison through the realistic channel.** We also compared these codes across the channels C111, C253, and MemSim introduced in Section 6.2, all of which have an overall channel error rate of $10.36\%$. Specifically, MemSim simulates the IDS channel derived from Nanopore sequencing.

The results are illustrated in Table 5. It can be observed that high-error-rate channels severely degrade the performance of compared codes, while the proposed THEA-Code outperforms them by a significant margin. Moreover, the compared codes, lacking the ability to adapt to specific channels, show a noticeable decline in performance as the channel transitions from the simpler C111/C253 to the more realistic MemSim. In contrast, THEA-Code leverages customized channel-specific designs, achieving the best performance on MemSim across all three channels.

### 6.4. More experiments in the Appendices

The gradients analysis of the differentiable IDS channel, with regard to the input sequence and the profile sequence, are presented in Appendix C. Ablation studies on the auxiliary reconstruction loss are presented in Appendix D, including the optimization of the loss weight $\mu$ and different auxiliary patterns. The complexity analysis and time consumption are presented in Appendix E.

## Impact Statement

This paper presents work whose goal is to advance the field of Machine Learning, Error-Correcting Code, and DNA Storage. There are many potential societal consequences of our work, none which we feel must be specifically highlighted here.

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

## A. Comparison Experiments

To evaluate the effectiveness of the proposed methods, we conducted comparison experiments against three prior works, which are:

- a combinatorial code that can correct single IDS errors over a 4-ary alphabet from Cai (Cai et al., 2021);

- a segmented method for correcting multiple IDS errors, called DNA-LM from (Yan et al., 2022);

- a well-known, efficient heuristic method, called HEDGES, from a DNA storage research (Press et al., 2020).

These methods typically offer only a few discrete, fixed configurations. We made efforts to align their settings as closely as possible. For Cai's combinatorial code, the code rates are fixed based on the code lengths. In our experiments on Cai, only the code rates are matched, with the code length determined according to the code rate[4]. For DNA-LM, we maintained the codeword length around 150, adjusting the number of segments to match code rates. For HEDGES, only binary library is publicly available, and it supports fixed code rates in $\{0.75, 0.6, 0.5, 1/3, 0.25, 1/6\}$. HEDGES' inner code was tested independently for comparison. We list all the source lengths $\ell_s$, codeword lengths $\ell_c$, and code rate $r$ used in the experiments in Table 7.

Table 7: The testing configurations for the comparison experiments. Each cell includes the code rate, message length, and code length. The settings are tried to be aligned, except the Cai configuration has a code length that does not align with 150, and HEDGES uses fixed code rates of $0.60$ and $0.75$, which are not aligned.

|  | $r_1 = \ell_{s1}/\ell_{c1}$ | $r_2 = \ell_{s2}/\ell_{c2}$ | $r_3 = \ell_{s3}/\ell_{c3}$ | $r_4 = \ell_{s4}/\ell_{c4}$ |
|---|---|---|---|---|
| Cai | 0.33=7/21 | 0.50=16/32 | 0.67=32/48 | 0.83=85/102 |
| DNA-LM | 0.34=50/148 | 0.51=77/152 | 0.68=96/142 | 0.84=124/148 |
| HEDGES | 0.34=52/155 | 0.50=76/152 | *0.60=92/153* | *0.75=115/155* |
| THEA-Code | 0.33=50/150 | 0.50=75/150 | 0.67=100/150 | 0.83=125/150 |

The experiments were conducted on the default IDS channel with $1\%$ error probability, as well as its variations, Asc and Des, introduced in Section 6.2. The results is illustrated in Figure 4. The experiments handled failed corrections by directly using the corrupted codeword as the decoded message.

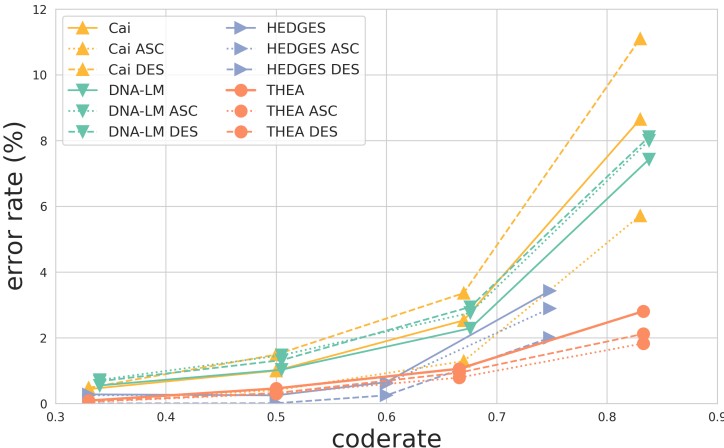

Figure 4: The error rates of the comparison experiments. Results for Cai, DNA-LM, HEDGES, and THEA-Code are shown across Hom, Asc, and Des channels, with respect to their code rates.

The results for Cai's method indicate that directly applying classical combinatorial codes to a $1\%$ IDS error probability channel with a codeword length of 150 is impractical. The observed error rates are high, even though these values

---

[4]It is important to note that code length plays a critical role in these experiments, as longer codewords are more likely to encounter multiple errors that cannot be corrected. Thus, Cai's performance here is just a baseline statistic of multi-errors with respect to the length, and performance may degrade with increased length.

were obtained with shorter code lengths than 150. The segmented method with sync markers in DNA-LM supports a codeword length of 150 and can correct multiple errors across different segments. However, it also exhibits a high error rate, indicating a nonnegligible likelihood of multi-errors occurring within the same segment. For HEDGES, while the results are commendable, the code rate is restricted to a limited set of fixed values. The results of THEA-Code demonstrate the effectiveness of the proposed method. At lower code rates, THEA-Code achieves a comparable error rate to HEDGES. At higher code rates, the proposed method outperforms HEDGES, achieving a lower error rate at a higher code rate, specifically $2.81\%$ error rate at $0.83$ code rate for THEA-Code v.s. $3.43\%$ error rate at $0.75$ code rate for HEDGES.

## B. Optimization of Hyperparameter Temperature $\tau$ in the Gumbel-Softmax Formula

To examine the impact of different temperature values $\tau$ in Equation (2), experiments were conducted with various settings of $\tau \in \{0.25, 0.5, 1, 2, 4, 8\}$. Since the disturbance-based discretization is designed to encourage greater discretization of the codeword, the codeword entropy $\mathcal{H}$, as defined in Equation (20), and the validation NER were tracked throughout the training phase

As shown in Figure 5, lower temperature ($\tau = 0.25$) has an effect in discretization, but result in unstable and poor model performance, while higher temperatures ($\tau \in \{2, 4, 8\}$) lead to both poor discretization and high NER.

## C. Gradients to the Differentiable IDS Channel

To investigate whether the simulated IDS channel back-propagates the gradient reasonably, the channel output $\hat{c} = \text{DIDS}(c)$ is modified by altering one base to produce $\hat{c}'$. The absolute values of the gradients of $\mathcal{L}(\hat{c}, \hat{c}')$ with respect to the input $c$ after back-propagation are presented in Figure 6. For instance, subfigure $\text{del}(+3)$ indicates that the IDS channel modifies $c$ to $\hat{c}$ by performing a deletion at index 0. The output $\hat{c}$ is then manually modified by applying a substitution at position $+3$. The gradients of $\mathcal{L}(\hat{c}, \hat{c}')$ with respect to $c$ are plotted over the window $[-2, +6]$.

It is suggested in Figure 6 that the proposed differentiable IDS channel back-propagates gradients reasonably. The gradients shift by one base to the left (resp. right) when the IDS channel performs an insertion (resp. deletion) on $c$. When the IDS channel operates $c$ with a substitution, the gradients stay at the same index. This behavior demonstrates that the channel is able to trace the gradients through the IDS operations. Specifically, in the case $\text{ins}(+0)$, the channel-inserted base in $\hat{c}$ at $\text{idx}$ is manually modified. As a result, no specific base in $c$ has a connection to the manually modified base, leading to a diminished gradient in this scenario.

### C.1. More on the gradients to differentiable IDS channel

Above, we illustrated that the differentiable IDS channel can effectively trace gradients through the IDS operations. In this section, we focus on evaluating the channel's capability to recover the error profile through gradient-based optimization.

Given a codeword $c$, an empty profile $p_0$ which defines the identity transformation of the IDS channel such that $\hat{c} = c = \text{DIDS}(c, p_0)$, and a modified codeword $\hat{c}'$ which is produced by manually modifying $c$ through an insertion, deletion, or substitution at position $\text{idx}$, the gradients of $\mathcal{L}(\hat{c}, \hat{c}')$ are computed with respect to both the input codeword $c$ and the empty profile $p_0$. The average gradients, calculated over 100 runs, are plotted in Figure 7 with position $\text{idx}$ aligned to 0.

In Figure 7, it is suggested that, when performing an insertion or deletion, the gradients with respect to the codeword are distributed after the error position $\text{idx}$. This aligns with the fact that synchronization errors (insertions or deletions) can be interpreted as successive substitutions starting from the error position, especially when the actual error profile is unknown. When performing a substitution, the gradients naturally concentrate at the error position $\text{idx}$.

Regarding the empty profile, $p_0 = 0$, the gradients also exhibit meaningful patterns. For an insertion, the substitution area after $\text{idx}$ is lighted by the gradients, supporting the view that an insertion can be seen as a sequence of substitutions if error constraints are absent. Additionally, the insertion area of the profile is also lighted, which makes sense since an insertion may also be interpreted as a series of substitutions followed by an ending insertion. For deletion errors, similar patterns are observed: the gradients are distributed in the areas of substitutions and deletions after the error position $\text{idx}$, since the deletion can also be viewed as a series of substitutions, or as several substitutions and an ending deletion. For substitution errors, the gradients again concentrate at the error position $\text{idx}$, as substitutions do not cause sequence mismatches.

Utilizing energy constraints on the profile may be helpful for specific profile applications. In this work, only the gradients

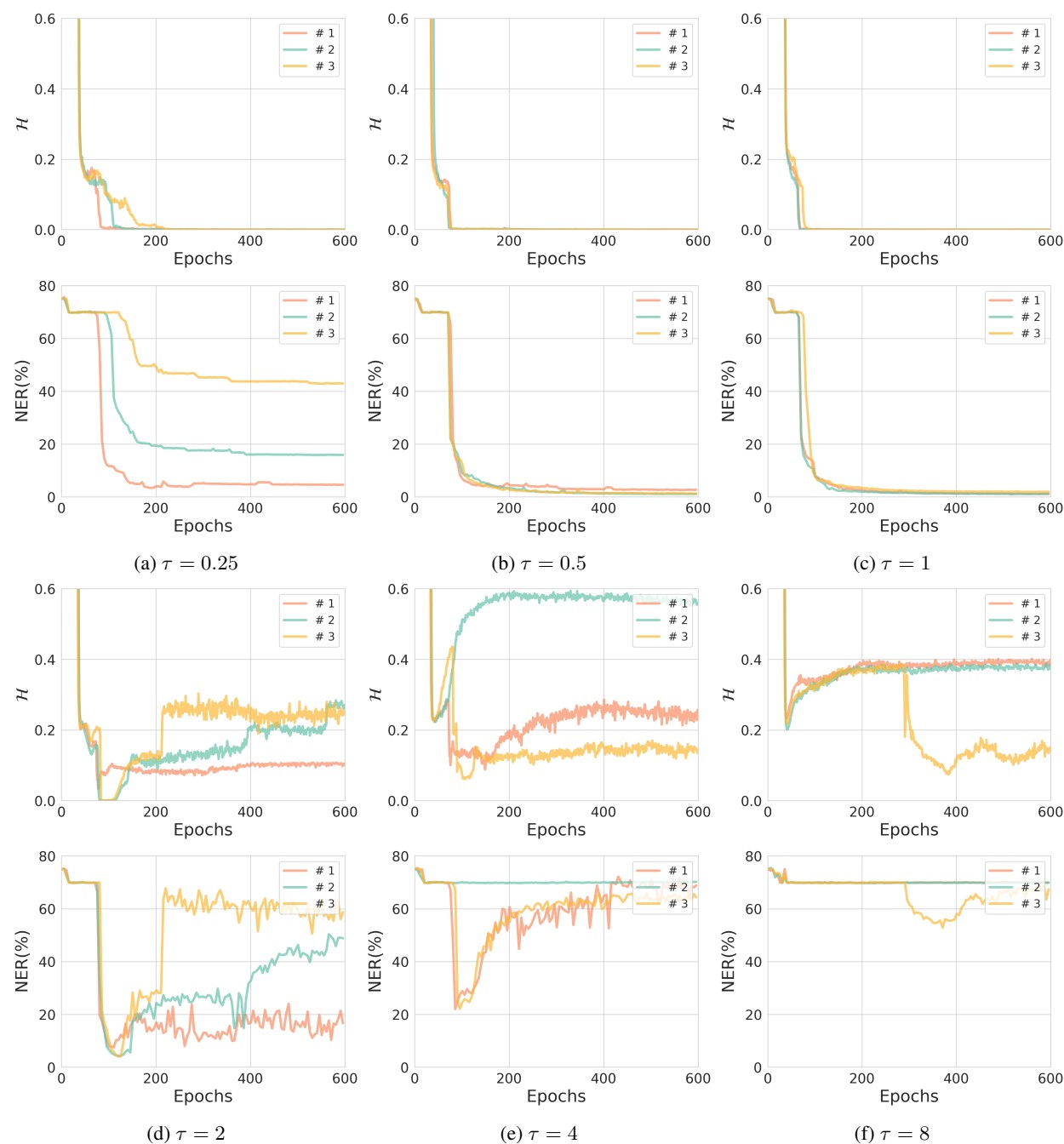

Figure 5: The codeword entropy $\mathcal{H}$ and the validation NER for various choices of $\tau \in \{0.25, 0.5, 1, 2, 4, 8\}$. Each curve in the subfigures represents one of the 3 runs conducted in the experiment and is plotted against the training epochs.

with respect to the codeword participate in the training phase, the existing version of the simulated differentiable IDS channel is assumed to be adequate.

## D. Ablation Study on the Auxiliary Reconstruction Loss

### D.1. Effects of the auxiliary reconstruction loss

Experiments with different choices of the hyperparameter $\mu$ were conducted, which are $\mu = 0$ indicating the absence of the auxiliary reconstruction loss, and $\mu \in \{0.5, 1, 1.5\}$ for different weights for the auxiliary loss. The validation NER and the

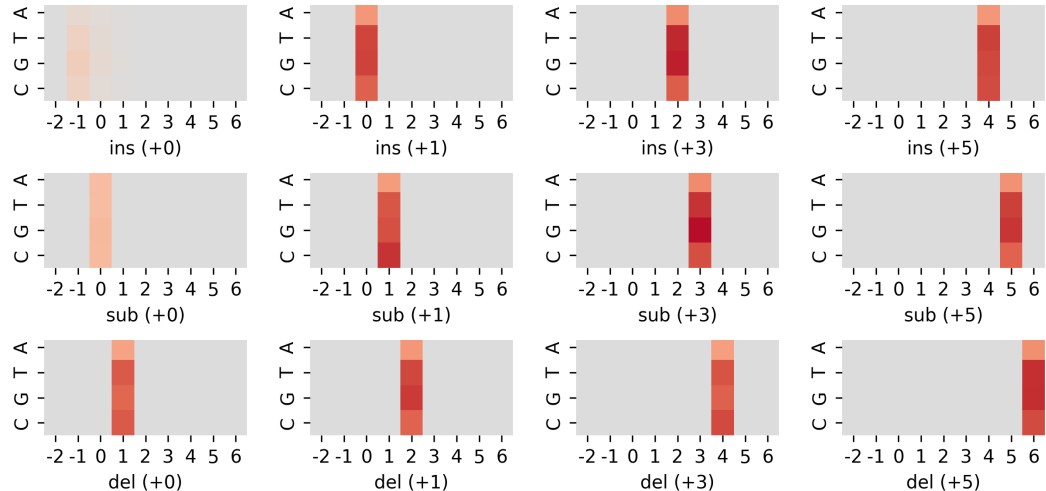

Figure 6: The averaged absolute gradients with respect to the input $c$ over 100 runs. The corresponding IDS operations were performed at an aligned $\text{index} = 0$ by the simulated differentiable IDS channel, the gradients were back-propagated from position $+k$ of the channel output $\hat{c}$. It is suggested that the gradients identify their corresponding position in the input: $+k-1$ for insertion, $+k$ for substitution, and $+k+1$ for deletion.

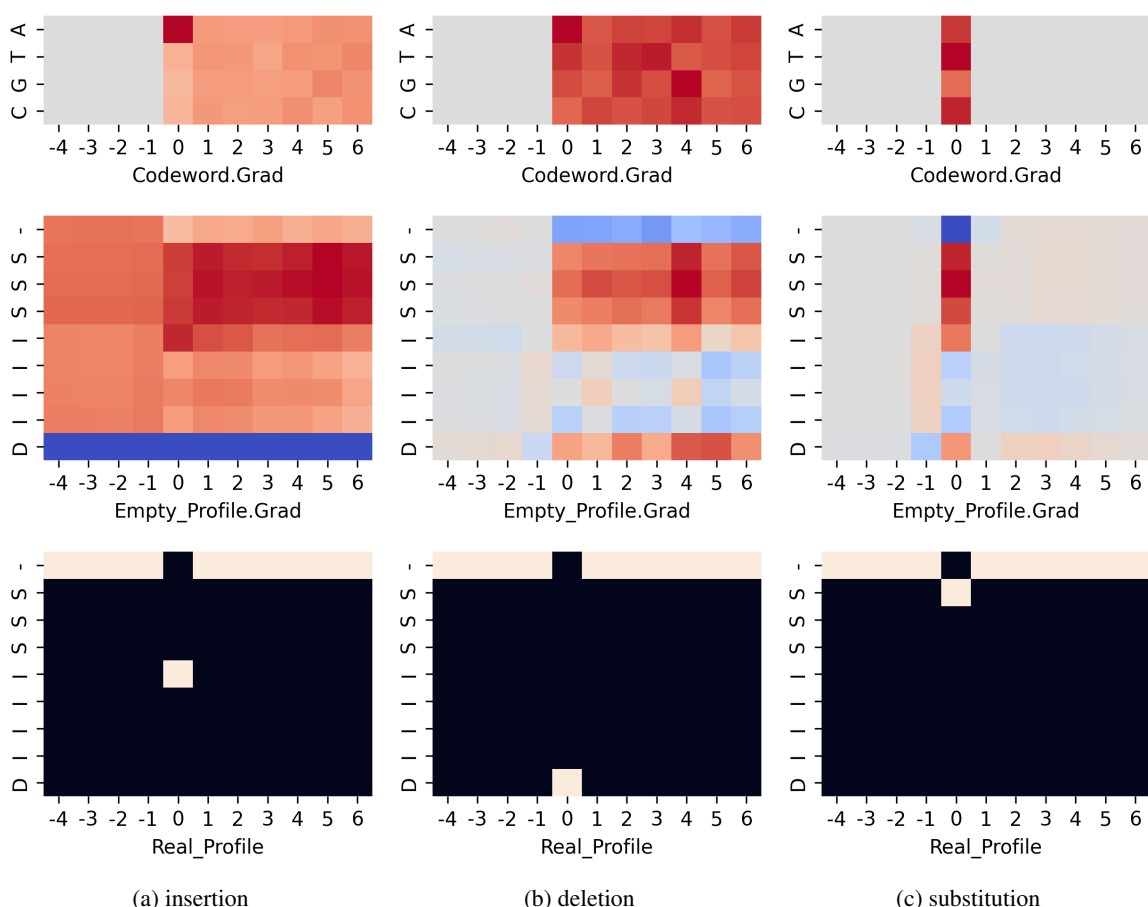

(a) insertion      (b) deletion      (c) substitution

Figure 7: The gradient distribution with respect to the input codeword and the empty profile, when the output codeword is manually modified. The figures display the averaged gradients over 100 runs, visualizing how the gradients were back-propagated in different cases of insertions, deletions, and substitutions in the output codeword.

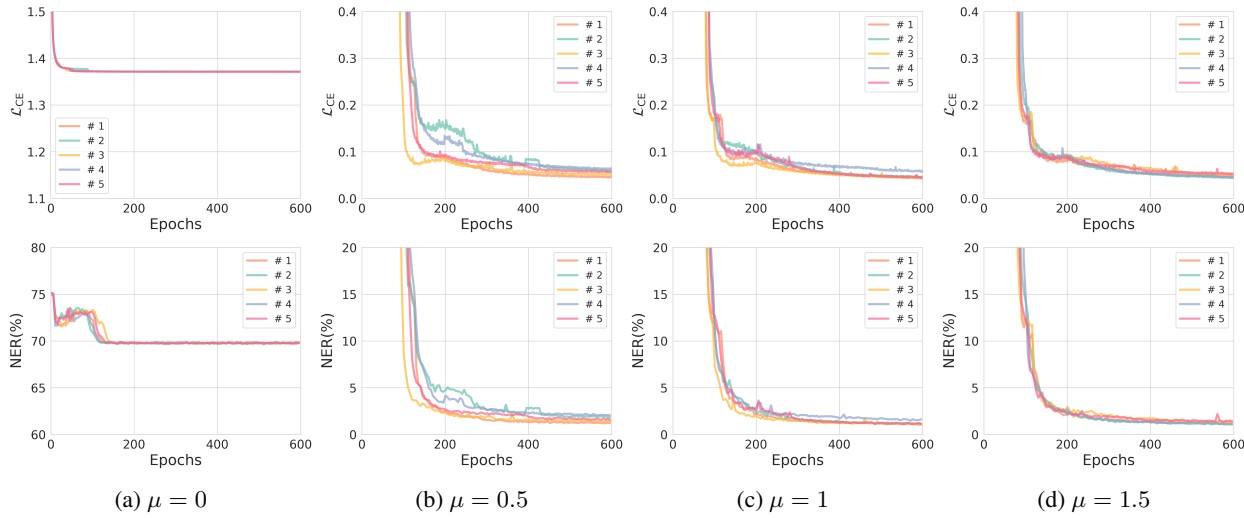

(a) $\mu = 0$  (b) $\mu = 0.5$  (c) $\mu = 1$  (d) $\mu = 1.5$

Figure 8: The reconstruction loss $\mathcal{L}_{\text{CE}}$ between the source and sink sequences, and the validation NER for various choices of $\mu \in \{0, 0.5, 1, 1.5\}$. Each curve in the subfigures represents one of the 5 runs conducted in the experiment and is plotted against the training epochs.

reconstruction loss between source and sink sequences are plotted against the training epochs.

The first column of Figure 8 indicates that without the auxiliary loss, all 5 runs of the training fail, producing random output. By comparing the first column with the other three, the effectiveness of introducing the auxiliary loss can be inferred. In the subfigures corresponding to $\mu \in \{0.5, 1, 1.5\}$, all the models converge well, and the NERs also exhibit a similar convergence. This suggests the application of the auxiliary loss is essential, but the weight of this loss has minimum influence on the final performance.

**D.2. Auxiliary loss on patterns beyond sequence reconstruction**

In Appendix D.1, the necessity of introducing a auxiliary reconstruction task to the encoder is verified. After these experiments, a natural question arises: How about imparting the encoder with higher initial logical ability through a more complicated task rather than replication? Motivated by this, we adopted commonly used operations from existing IDS-correcting codes and attempted to recover the sequence from these operations using the encoder. In practice, we employed the forward difference $\text{Diff}(\boldsymbol{s})$, where

$$\text{Diff}(\boldsymbol{s})_i = s_i - s_{i+1} \mod 4, \tag{21}$$

the position information-encoded sequence $\text{Pos}(\boldsymbol{s})$, where

$$\text{Pos}(\boldsymbol{s})_i = s_i + i \mod 4, \tag{22}$$

and their combinations as the reconstructed sequences.

The evaluation NERs against training epochs are plotted in Figure 9 under different combinations of the identity mapping I, Diff, and Pos. It is clear that the reconstruction of the identity mapping I outperforms Diff and Pos. Introducing the identity mapping I to Diff and Pos helps improving the convergence of the model, but final results have illustrated that they are still worse than simple applying the identify mapping I as the auxiliary task. These variations may be attributed to the capabilities of the transformers in our setting or the disordered implicit timings during training.

# E. Transformer, Complexity, and Time Consumption

Transformers (Vaswani et al., 2017), well-known deep learning architectures, rely on the attention mechanism. Each head of a Transformer model processes features according to the following formula:

$$\text{Attention}(Q, K, V) = \text{softmax}\left(\frac{QK^T}{\sqrt{d_k}}\right) V. \tag{23}$$

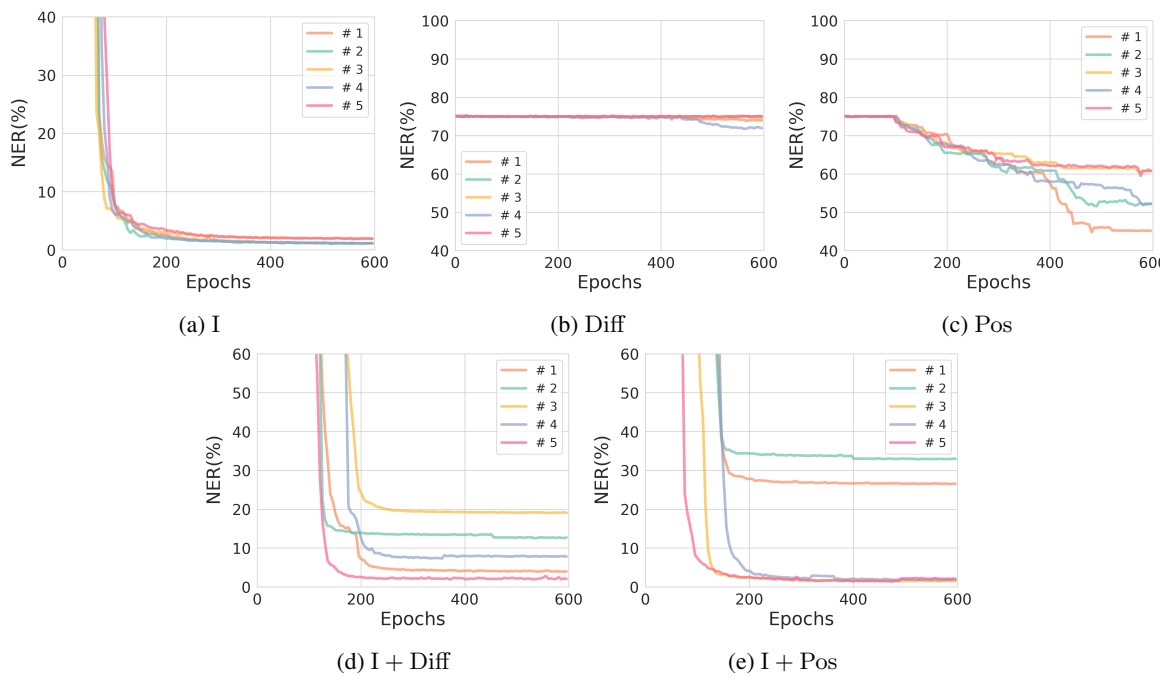

(a) I  (b) Diff  (c) Pos

(d) I + Diff  (e) I + Pos

Figure 9: The validation NER against the training epochs with different choices of auxiliary reconstruction. The reconstructed sequences are produced by combinations of the identity mapping I, Diff, and Pos, where + denotes sequence concatenating.

In this work, each layer comprises 16 attention heads with an embedding dimension 512, and a total of $3 + 3$ attention layers are used for the sequence-to-sequence model. Both the encoder and decoder are implemented as such sequence-to-sequence models. For the differentiable IDS channel, a $1 + 1$ layered sequence-to-sequence model is employed.

Since attention is calculated globally over the sequence in Equation (23), it has a complexity of $O(n^2)$. Without delving into the many efficient transformer architectures, the time consumption was measured by decoding $1,280,000$ codewords using an RTX3090. The encoder, which shares the same structure, exhibits similar performance. The results are acceptable and are presented in Table 8.

Table 8: Time consumption of decoding $1,280,000$ codewords for each source length $\ell_s$ by an RTX3090.

|          | $\ell_s = 50$ | $\ell_s = 75$ | $\ell_s = 100$ | $\ell_s = 125$ |
|----------|---------------|---------------|----------------|----------------|
| time (s) | 521.94        | 573.87        | 623.92         | 687.76         |