# OpenReview forum: "Disturbance-based Discretization, Differentiable IDS Channel, and an IDS-Correcting Code for DNA Storage"
_ICML.cc/2025/Conference — Submitted to ICML 2025_

### Official Review · Reviewer_PNwD · 2025-03-11

**Overall Recommendation:** 3

**Summary:**

The authors propose THEA-Code, a IDS-correcting code for storing DNA, where the codes are subject tor insertion, deletion, and substitution errors. Their approach has two main components: first, they train a differentiable model to simulate the IDS channel. Using the trained channel, they additionally train an auto-encoder with Gumbel-Softmax discretization, which is able to reconstruct the DNA, even when the codewords are corrupted.

**Claims And Evidence:**

1. The authors claim "commendable performance" of their approach. This is supported by their experiments showing consistent improvements over two pieces of previous work in Table 5.

2. The authors claim their method works in realistic settings, and this was demonstrated using a simulated channel called MemSim. However, this remains a simulated setting, and I am not sure if a more realistic setting is available.

3. The authors claim Gumbel softmax is much better than vanilla softmax, and provided an ablation study in Figure 3. Here, the authors show Gumbel softmax produce lower entropy, which corresponds to better performance. Here, I am not sure if simply adjusting the temperature of the softmax (which in turn reduces entropy) will achieve the same thing.

**Essential References Not Discussed:**

N/A.

**Experimental Designs Or Analyses:**

I checked the experiments in the main paper.

**Methods And Evaluation Criteria:**

1. The authors only use NER as the metric, which makes sense in this case. However, I am not sure if there are common and more advanced metrics in this area.

2. The authors experiment with a variety of channels at different code rates, which shows the method's robustness.

**Other Comments Or Suggestions:**

N/A.

**Other Strengths And Weaknesses:**

Strengths

1. The proposed method significantly outperform previous methods.
2. The authors conduct ablation studies for Gumbel Softmax, as well as testing their approach in various setups.
3. The authors provide detailed analyses in their Appendix.

Weaknesses:

1. The intuition for Gumbel Softmax is a bit unclear to me (I might be missing something). The authors explain that they adopted Gumbel Softmax so that it will "constrain the logits x to produce one-hot-like probabilitiy vectors" (also shown through their theorem). However, if the goal is to simply produce sharp distributions, this can be achieved by adjusting the softmax temperature.
2. Lacking an ablation study and detailed analysis for the simulated IDS channel. It would interesting to see how the system performs without it, and how accurate the simulated IDS channel is. (Correct me if I missed these.)
3. The error profile seems limiting. The simulated IDS channel takes an error profile vector, which is a simple statistics over types of errors encountered. However, it is unclear how a simple vector can summarize more complex errors.

**Questions For Authors:**

N/A.

**Relation To Broader Scientific Literature:**

The authors argue that previous work also uses auto-encoder-based codes (Baldi et al.); however, they do not have the simulated IDS channel, which make it harder to take advantage of the specific error profile.

**Theoretical Claims:**

No.

---

> ### Author Rebuttal · Authors · 2025-03-29
>
> **We sincerely thank the reviewer for their valuable efforts. We will revise the manuscript accordingly. We hope our rebuttal has addressed the concerns.**
>
> **Q1**: Is a more realistic setting than MemSim available?
>
> **A1**: Firstly, a simulated channel is essential for training such a model, as alternating between training epochs and wet-lab experiments is neither time- nor resource-efficient.
>
> As far as we know, MemSim is currently a most advanced option available. It utilizes base-context-related statistics to simulate the biochemical process. In reality, DNA storage channels vary significantly depending on the specific methods, equipment, etc. As a result, users may prefer to train codes tailored to their own channels rather than relying on a universal simulated channel. This is also why we present our work as a generalizable method rather than just a standalone model.
>
> In future research, a generative model like VAE, in combination with the differentiable IDS channel, could be explored for a NN-based simulation.
>
> **Q2**: If the goal is to simply produce sharp distributions, will simply adjusting the temperature of the softmax will achieve the same thing?.
>
> **A2**: This is an insightful thought. Our initial attempt involved adjusting the softmax temperature, but this alone was not the optimal choice. In training phase, we want to progressively sharpen the encoder’s output distribution while ensuring the decoder remains sensitive only to the maximum entry of the distribution.
>
> Applying low-temperature softmax in training may hinder convergence. If the autoencoder has not yet learned meaningful features, an overly sharp distribution can prevent it from reaching a non-trivial solution. Experiments across different settings showed that while convergence is possible, it requires careful tuning of $t$, making training less robust. Some results under the default setting are listed below:
>
> |  | Gumbel Softmax | Softmax t=0.1 | t=0.2 |t=0.3| t=0.4 | t=0.6 |
> |---:|:---:|:---:|:---:|:---:|:---:|:---:|
> | NER | 1.06 | 6.10 | 1.54 | 3.91 | 1.89 | 21.00 |
> | Entropy | 2e-5 | 4e-4 | 2e-3 | 2e-4 | 1e-3 | 0.08 |
>
> It suggests that adjusting the softmax temperature can help generate low-entropy codewords, but the functionality of the codewords is compromised compared to the proposed disturbance constraints.
>
> Beyond applying fixed low-$t$ softmax, we also explored alternative approaches, including: random $t$ softmax, $\sin t$ softmax, and optimizing an entropy constraint on the distributions. Among these, the entropy constraint also worked, but similar to adjusting the softmax $t$, its weight $\lambda$ required careful tuning to balance distribution sharpening, model convergence, and preventing the decoder from exploiting soft distributions.
>
> **Q3**: It would be interesting to see how the system performs without the simulated IDS channel, and how accurate the simulated IDS channel is.
>
> **A3**:
> We would like to thank the reviewer for this insightful comment.
>
> Without the simulated channel, the entire framework would not function, as the neural-network-based channel is essential for back-propagating gradients to the encoder. If we bypass the IDS channel using a straight-through approach, the task degenerates into a trivial copy-and-paste operation with 100% accuracy. This has been confirmed in unreported experiments by setting the channel error rate to 0.
>
> We evaluated the accuracy of the simulated IDS channel (DIDS) by comparing it to the groundtruth produced by the conventional IDS channel (CIDS). Part of the results are as follows:
>
> | CH Err | 0 | 1% | 5% | 10% | 20% | 30% | 40% |
> |---:|:---:|:---:|:---:|:---:|:---:|:---:|:---:|
> | CIDS==DIDS | 100 | 99.8 | 99.4 | 99.1 | 94.2 | 66.6 | 41.5 |
>
> We found that the learned channel is reliable for simulating channels with an error rate of less than 20%.
>
> We will append a section in App C to include detailed results from these experiments.
>
> **Q4**: The simulated IDS channel takes an error profile vector, which is a simple statistics over types of errors encountered. However, it is unclear how a simple vector can summarize more complex errors.
>
> **A4**: The error profile records the errors that occur in a sequence. For instance, given ATGGC and an error profile of (Ins C, Ins T, 0, Del, 0,0,0), the resulting sequence would be **CT**A~~T~~GGC. This strategy covers all possible error types.
>
> We will add an App Sec on how the profile is defined.
>
> In the simulated channel the profile is DNA-depended. The differentiable IDS channel faithfully transforms the sequence according to the error profile. Thus, the channel is fully defined by how the profile vector is generated. Context-free channels, such as C111, generate error profiles based on preset probabilities, while MemSim is sequence-dependent, generating profiles based on DNA sequences by sampling from $P(profile|DNA)$. The right column of Line 371 in the manuscript describes this part in detail.

---

### Official Review · Reviewer_7v5k · 2025-03-11

**Overall Recommendation:** 3

**Summary:**

They proposed a universal method for designing tailored IDS-correcting codes across varying channel settings.
1. They propsed a disturbance-based discretizationto discretize the features of the autoencoder, which applies a Gumbel SoftMax to code the the alphabet {A, T, G, C}.
2. A simulated differentiable IDS channel is developed as a differentiable alternative for IDS operations,  which is the key to address IDS or DNA-related problems using deep learning methods.

**Claims And Evidence:**

convincing

**Essential References Not Discussed:**

N/A

**Experimental Designs Or Analyses:**

soundness

**Methods And Evaluation Criteria:**

make sense

**Other Comments Or Suggestions:**

1. I suggest moving Figure 2 to page 5 for easy viewing. In addition, Figure 2 lacks explanation for some belonging, such as "sink".
2. Authors should consider showing the structure of the model in detail, even if the author provides the code.
3. The modeling approach in the field of ECC is similar; however, can you explain why conventional approaches do not address IDS-correcting codes across varying channel settings? The author should add context and meaning to this section.
4. The link to the code provided by the author seems to be cancelled and I can't view it.

**Other Strengths And Weaknesses:**

First of all, I'm not an expert in DNA storage, but I applaud the author's contribution to using the Transformer model in this area.

Strength

1. The article is readable, even for people unfamiliar with the topic.
2. This is a valuable field, and this work is the first to model it using Transformers.

Weaknesses

1. This paper lacks a clear benchmark, including dataset Settings and error type distribution Settings. The author should explain how the dataset was constructed in the experiments section so that the researcher can follow along.
2. Although the field of DNA storage lacks corresponding benchmarks, authors should consider comparing with similar methods in the field of ECC:
[1] Choukroun, Yoni, and Lior Wolf. "Error correction code transformer." Advances in Neural Information Processing Systems 35 (2022): 38695-38705.
[2] Wang, Hanrui, et al. "Transformer-QEC: quantum error correction code decoding with transferable transformers." arxiv preprint arxiv:2311.16082 (2023).

**Questions For Authors:**

1. As the proposed model is designed to handle source sequences and codewords of constant lengths, is it possible to process a short one with paddings?
2. line 238:"Particularly, when imposing constraints to enforce greater discreteness in the codeword, the joint training of the encoder and decoder resembles a chicken-and-egg dilemma, where the optimization of each relies on the other during the training phase." is not clear, could you provide more details?
3. I noticed that the dataset is a randomly generated sequence, however the authors did not specify the exact rules. Is any sequence allowed, and how is an error operation defined? I expect the authors to state the definition of allowed sequences as well as visualizations in the previous sections.
4. When using both combinatorial codes for correcting a single error and a burst of errors, is the proposed method competitive?

**Relation To Broader Scientific Literature:**

DNA storage, ECC

**Theoretical Claims:**

Probably correct

---

> ### Author Rebuttal · Authors · 2025-03-25
>
> **We sincerely thank the reviewer for their valuable comments. We hope our rebuttal has adequately addressed the concerns. Minor concerns not mentioned will also be revised.**
>
> **Q1**: The code is empty.
>
> **A1**: This appears to be a cache issue with the anonymous hosting platform, as several similar cases have also been reported on their GitHub page. It has now been fixed and is accessible at the same link: https://anonymous.4open.science/r/THEACode .
>
> Nonetheless, it’s our fault for not thoroughly verifying the availability of the code.
>
> **Q2**: How the dataset/profile was constructed and error operation defined?
>
> **A2**: The DNA sequences are randomly generated with equal probabilities for ATGC, no inherent patterns. The error profiles are constructed according to the respective channel settings. For instance, under the default setting, each profile position undergoes an Ins, Del, or Sub with equal probability (Err_Rate/3), and Ins/Sub are further distributed equally among bases.
>
> For the simulated realistic channel MemSim, we use its official implementation to generate the output sequence $s’$ from the input $s$, then infer an error profile $p(s,s’)$.
>
> We will add an App Sec for the error profile. Here's a brief example: given the sequence ATGGC and an error profile of (Ins C, Ins T, 0, Del, 0,0,0), the resulting sequence would be **CT**A~~T~~GGC.
>
> **Q3**: Comparing with similar NN-based ECC.
>
> **A3**: We follow the research trend of NN-based ECCs, which focus on linear codes such as LDPC. However, correcting errors in AWGN channels is fundamentally different from handling IDS errors. Ins and Del shift the entire sequence, making them inherently unsuitable for linear codes.
>
> Transplanting these methods to IDS correction would likely face the same challenges addressed in this manuscript. On the other hand, directly applying such approaches to IDS errors would not differ significantly from using conventional linear codes, which has been explored in very early DNA storage research with unsatisfactory results.
>
> **Q4**: Why conventional approaches do not address varying channel settings?
>
> **A4**: We believe this is due to two main reasons:
>
> + In conventional ECC research, handling complex AWGN channels has not been a primary focus, as such complexity is less critical than in DNA storage. As evidence, although NN-based ECCs offer advantages in complex channels, most existing works do not emphasize this capability.
> + IDS correction follows a different approach from conventional ECCs. Even for the simplest case of correcting a single IDS error, the mathematical foundations remain open, and an optimal code has yet to be established. Designing more advanced IDS codes for complex channels thus remains a challenging open problem.
>
> **Q5**: Is it possible to process a short codewords with paddings?
>
> **A5**: Padding is not explicitly described but is actually used throughout the work. This is necessary because synchronization errors (Ins/Del) alter the sequence length.
>
> Applying variable-length sequences is an interesting topic. It raises the question of whether the model would learn individual codes for different lengths or a consistent code that accommodate both short and long sequences.
>
> This model was trained with fixed-length sequences, as sequence length is a key prior knowledge for correcting synchronization errors. For example, without the knowledge of codeword length, a broken codeword of length n could originate from either a length n-1 codeword with an Ins, a length n codeword with Subs, or a length n+1 codeword with a Del. Multiple errors would further complicate this scenario. We infer that variable-length sequences would increase the task's difficulty significantly.
>
> Consider this, it may require extensive research in future work to answer this question.
>
> **Q6**: More details on Chicken-and-Egg dilemma.
>
> **A6**: The dilemma arises from the interdependence of the encoder and decoder during training. Specifically, when using discreteness constraints, if the encoder converges prematurely to a local minimum due to these constraints, the entire framework fails to function properly. To mitigate this, we introduce the auxiliary task, which serves as a logical "warm-up" for the encoder. This task is simple yet effective, as in App D.
>
> **Q7**: Competitive to combinatorial codes in correcting single error?
>
> **A7**: The accuracy cannot surpass combinatorial codes in this scenario, as they are mathematically guaranteed to correct a single error. However, from our other research efforts, it’s found that a NN-based decoder can 100% decode the combinatorial codewords.
>
> To evaluate whether the end-to-end method competitive to combinatorial code in single IDS channel, we conducted experiments with a code rate aligned to Cai’s code at 34/50 and 133/150. The reported **NER is: 1.6%, 2.1%**, respectively, inferior to the combinatorial code, although the THEA-Code performs far better in correcting multiple errors.

---

### Official Review · Reviewer_g35h · 2025-03-15

**Overall Recommendation:** 2

**Summary:**

This work proposes THEA-code, an auto encoder for learning IDS-correcting codes. It does this in two stages: (1) learning a differentiable IDS channel from a ATGC sequences from CIDS, and then (2) using the learned IDS channel to train an auto encoder to automatically learn a IDS-correcting code.

**Claims And Evidence:**

The claims are supported with experimental results.

**Essential References Not Discussed:**

To the best of my knowledge, no.

**Experimental Designs Or Analyses:**

The experimental results seem overall sound and valid. One concern I had was the codeword length is always fixed to 150. Why not make the model's rate fixed and instead output codewords that are length $\ell_s$/(code rate)? This would seem more useful in practice. Also, it raises the concern that the model is overfitted on a very specific code worth length. Is the model able to generalize to different codeword lengths? i.e., if I wanted to use a rate of 0.50 but a source sequence of 300? This also raises a question of how the comparison methods (DNA-LM, Cai, and HEDGES) operate. Is the comparison across all methods done using the same exact source sequence(s), resulting in the the same length codeword length for all methods (at a fixed code rate), and then comparing the error rates? If not, I believe there may be slight unfairness in the comparison. In any case, this should be mentioned.

**Methods And Evaluation Criteria:**

Mostly. One question I had is whether any real genomic datasets are used, or if random AGTC sequences are drawn for the experimental sections. I am not an expert on genomic data, so I do not know if the AGTC sequences may have memory or can be assumed iid. To be more convincing, I think the experiments should include error correction performance benchmarks on real-world genomic data.

**Other Comments Or Suggestions:**

I wonder if some sort of "adversarial" training could be done, where the learned channel and error correcting code are adversaries, and they are both learned from scratch. This would alleviate the need to pretrain the IDS channel on IDS channel inputs and outputs.

**Other Strengths And Weaknesses:**

Aside from what was mentioned above, I think the paper is well-written and easy to follow. Some other recommendations I would have are:

- putting some experiments regarding the accuracy of the learned IDS channel in the main text
- including a diagram of some of the IDS operations when discussing the differentiable channel in section 4. Showing how the probability vector representations are represented throughout the DIDS and CIDS with a working example (say, for insertion and deletion) would be very helpful to the reader.

My main concern with the paper is Theorem 3.1 (see above). I also feel it is a bit disconnected to the rest of the paper, as it is never discussed later on in the experiments (i.e., to verify that sparsity is achieved). The core contributions seem to really be in learning the channel, and then using the differentiable channel to learn the code.

**Questions For Authors:**

Please see the above.

**Relation To Broader Scientific Literature:**

I am not an expert in DNA coding. However, the approach seems novel (using the 3-simplex to represent soft versions of the ACTG symbols, and then learning transformer-based models for both the channel and the error correcting code). The authors have included a fairly extensive literature review for deep learning applied to coding theory, which I believe is the closest area of research to this work.

**Theoretical Claims:**

I think the theoretical result (Theorem 3.1) needs some work. The theorem statement itself is vague and could benefit from a precise mathematical description. This would help the reader understand what is being proved in the proof. The current theorem statement reads more like a remark. Regarding the proof I do not know if both the $\epsilon_1, \epsilon_2$ in the proof are small (as it says that either $y_1$ or $y_2$ is less than $\epsilon$, but the final bound on $\pi_1$ is in terms of both $\epsilon_1$ and $\epsilon_2$), so it is hard to tell whether sparsity is achieved. Perhaps the theorem statement would make sense if it relates the sparsity of $\pi$ to the convergence tolerance $\epsilon$.

Additionally, there is no mention of a full proof anywhere for general $\tau$ and more than 2 logits.

---

> ### Author Rebuttal · Authors · 2025-03-25
>
> **We sincerely thank the insightful feedback, which is invaluable in improving our manuscript. As this is the only negative score, we genuinely hope our rebuttal has addressed all the concerns and that the reviewer may reconsider the score.**
>
> **Q1**: The code is empty.
>
> **A1**:
> This appears to be a cache issue with the anonymous hosting platform, as several similar cases have also been reported on their GitHub page. It has now been fixed and is accessible at the same link: https://anonymous.4open.science/r/THEACode .
>
> Nonetheless, it’s our fault for not thoroughly verifying the availability of the code.
>
> **Q2**: Any genomic datasets are used? Random AGTC sequences?
>
> **A2**:
> We use random AGTC sequences, as DNA molecules serve as a memoryless medium for storing arbitrary information in DNA-based information storage.
>
> The use of genome-style or bio-compatible DNA sequences for **in vivo** storage has been explored (see [1]). However, in this **in vitro** storage, where DNAs exist in dry powder form, sequence constraints are relaxed. Typically, only patterns that are difficult to synthesize or sequence should be noticed, which is actually a property of the IDS channel (i.e., patterns introducing higher error rates) and should be in charge of channel modeling.
>
> When storing genomic information in DNA molecules, genomic knowledge is also unnecessary. The data is usually compressed **in silico** before storage, maximizing entropy and eliminating inherent sequence patterns.
>
> [1] An artificial chromosome for data storage, NSR
>
> **Q3**: Thm 3.1 needs some work. Proof for general $\tau$ and $n$ logits.
>
> **A3**:
> We will revise Thm 3.1 and its proof for clarity. Specifically, $\epsilon_1$ is similarly to the convergence tolerance, while $\epsilon_2$ accounts for the chance that $y_1$, as a sample from a distribution, deviates beyond this tolerance.
>
> Additionally, we will **provide a full proof** in the Appendix for general $\tau$ and $n$ logits, in about 1.5 extra pages. The proof follows the sketchy proof but includes additional details and a trick involving the mean value theorem for multivariable functions.
>
> **Q4**: Codeword length, why not fix source length? Still work at 300/600?
>
> **A4**:
> In DNA storage, the sweet point of molecule length is around 150 due to biochemical limitations. Shorter lengths require additional indexing resources, while longer lengths are currently neither time- nor cost-effective; excessively long synthesized DNA sequences accumulate high error rates. Therefore, we conducted experiments by fixing the codeword length rather than fixing the source length.
>
> As suggested, we had experiments with both shorter and longer codeword lengths at settings 25/50 and 300/600. Under a 1% error channel, training the code for 25/50 was much easier, while training for length 600 was relatively challenging and resulted in an inferior NER, as in
> ||25/50|75/100|300/600|
> |-|:-:|:-:|:-:|
> |NER|0.37|0.46|5.16|
>
> We acknowledge that the proposed method is not applicable to arbitrarily long sequences, as it relies on a plain Transformer, which is computationally impractical for very long inputs.
> However, encoding long DNA is not a currently urgent priority. Future work may explore more efficient Transformer variants for this purpose.
>
> **Q5**: Fairness of comparison. Source/Codeword length settings?
>
> **A5**:
> Yes, the comparison is not entirely fair, primarily because the compared methods use discrete settings. The compared source/codeword lengths are presented in Tab 7 App A. The most unfair setting is for Cai’s code, which uses smaller codeword lengths to align the code rate. However, in this case, Cai's accuracy is overrated rather than underestimated. This code is reliable for correcting single error but fails to correct multiple errors. Shorter codeword reduces the likelihood of encountering multiple errors in a channel with fixed error rates.
>
> **Q6**: Accuracy of the learned IDS channel.
>
> **A6**:
> We would like to thank the reviewer for this insightful comment. We evaluated the accuracy of the learned IDS channel (DIDS) by comparing it to the groundtruth produced by the conventional IDS channel (CIDS). Part of the results are:
>
> |CH Err|0%|1%|5%|10%|20%|30%|40%|
> |-:|:-:|:-:|:-:|:-:|:-:|:-:|:-:|
> |CIDS==DIDS|100.0|99.8|99.4|99.1|94.2|66.6|41.5|
>
> It suggests that the learned channel is reliable for simulating channels with error rate less than 20%.
>
> We will include the detailed results from these experiments.
>
> **Q7**: Theorem 3.1 is a bit disconnected to the rest of the paper, as sparsity is not discussed later on in the experiments.
>
> **A7**:
> The entropy of codewords, which directly reflects sparsity or discreteness, was recorded in Sec 6.1 and App B to illustrate the effect of Thm 3.1. The reviewer may have missed these parts, which is our fault due to the small text used after zooming out the figures for page limit. We will revise this in the updated version.
>
> **All other suggestions will be taken into account.**

---

### Official Review · Reviewer_3UV6 · 2025-03-15

**Overall Recommendation:** 3

**Summary:**

This paper presents THEA-code, an end-to-end autoencoder-based model for an IDS-correcting code. Extensive experiments demonstrate that THEA-Code can adapt effectively to various IDS channel conditions and outperforms existing IDS-correcting codes on simulated and realistic DNA storage channels. THEA-Code especially reduces error rates for realistic DNA storage channel conditions.

**Claims And Evidence:**

Yes; to the best of my knowledge, there are not any problematic claims.

**Essential References Not Discussed:**

I don't think any essential references are missing. However, I think readers unfamiliar with the area might feel there is a slight lack of background on DNA storage, IDS, etc. and what the model task is in the introduction.

**Experimental Designs Or Analyses:**

The autoencoder design and adaptation for the task seemed valid.

**Methods And Evaluation Criteria:**

Yes; nucleobase error rate (NER) was evaluated on C111, C253, and MemSim across multiple prior methods and THEA-code.

**Other Comments Or Suggestions:**

Page 2 has the typo "distrubance" instead of "disturbance."
Page 7 has the typo "apperent" instead of "apparent."

**Other Strengths And Weaknesses:**

Given that THEA-code appears to be the first end-to-end autoencoder framework for this task, along with extensive explanations it seems like there is sufficient originality and

**Questions For Authors:**

N/A

**Relation To Broader Scientific Literature:**

The use of a deep learning-based autoencoder mainly distinguishes this paper from previous work.

**Theoretical Claims:**

The theorems presented seemed correct, although sometimes hard to follow if all variables were not explicitly defined or explained in the previous text.

---

> ### Author Rebuttal · Authors · 2025-03-28
>
> **We sincerely thank the reviewer for their valuable efforts. We will revise the manuscript accordingly. We hope our rebuttal has addressed the concerns.**
>
> **Q1**: The theorem is sometimes hard to follow.
>
> **A1**:
> We will revise the main text of Theorem 3.1 for clarity.
>
> Additionally, we will provide a full proof in the Appendix for general $\tau$ and $n$ logits, in about 1.5 extra pages. The full proof follows the structure of the sketchy proof but includes additional details and a trick involving the mean value theorem for multivariable functions.
>
> **Q2**: Readers unfamiliar with the area might feel there is a slight lack of background on DNA storage.
>
> **A2**: We agree with the reviewer’s concern. Initially, we had such a paragraph on introducing DNA storage before the second paragraph in the introduction, but it was removed due to page limits. We will reintroduce this paragraph explaining the DNA-based information storage pipeline to provide better background for readers unfamiliar with the area.
>
> **Q3**: typos such as "distrubance", "apperent", etc.
>
> **A3**: We apologize for these typos and will correct them in the revised version. Additionally, we will conduct a thorough review of the manuscript.

---

### Decision · Program_Chairs · 2025-05-01

**Decision:**

Reject

**Comment:**

The authors introduce THEA-code, an autoencoder-based approach for generating Insertion, Deletion, and Substitution (IDS) error-correcting codes tailored for varying channel settings, particularly for DNA storage. Their framework includes a disturbance-based discretization method for the autoencoder features and a differentiable IDS channel as a substitute for non-differentiable IDS operations. This allows the autoencoder to converge and produce channel-customized IDS-correcting codes. Experiments demonstrate that THEA-code performs well across complex IDS channels, including realistic DNA storage channels, outperforming existing methods. Reviewers commend the application of a deep learning-based autoencoder to this problem and find the approach novel. The paper is also noted for being readable even for those unfamiliar with DNA storage.

Reviewers identify several areas for improvement. One reviewer finds the theoretical result (Theorem 3.1) vague and disconnected from the experiments, with concerns about the proof and the lack of discussion on sparsity in the experimental section. The paper lacks a clear benchmark with detailed dataset settings and error type distribution. Some reviewers feel the intuition for Gumbel Softmax is unclear, questioning if simply adjusting the softmax temperature could achieve similar results. The codeword length is consistently fixed, raising concerns about potential overfitting and generalizability to different lengths. The fairness of the comparison with existing discrete methods is questioned due to differing source/codeword lengths. The paper also lacks an ablation study and detailed analysis of the simulated IDS channel's accuracy.

Overall, the paper seems pretty good, but there is not enough support for acceptance, and hence I recommend rejection.

The reviewers offers several ideas for improvement before resubmission:
- Improve theoretical results and connect to practice
- Include a clearer benchmark description, detailing dataset construction, error type distribution, and evaluation settings.
- Investigate the model's performance with variable codeword lengths or fixed code rates and discuss its generalizability.
- Include experiments and analysis on the accuracy of the learned IDS channel compared to the conventional channel.
- Consider comparing the proposed method with other neural network-based error-correcting codes, even if they are designed for different channel types.